# A highly potent and safe pyrrolopyridine-based allosteric HIV-1 integrase inhibitor targeting host LEDGF/p75-integrase interaction site

**Tatsuya Maehigashi**[1☯], **Seohyun Ahn**[2☯], **Uk-Il Kim**[2☯], **Jared Lindenberger**[3☯],
**Adrian Oo**[1☯], **Pratibha C. Koneru**[3], **Bijan Mahboubi**[1], **Alan N. Engelman**[4,5],
**Mamuka Kvaratskhelia**[3]*, **Kyungjin Kim**[2]*, **Baek Kim**[1,6]*

1 Department of Pediatrics, School of Medicine, Emory University, Atlanta, Georgia, United States of America, 2 ST Pharm Co., Ltd., Seoul, South Korea, 3 Division of Infectious Diseases, University of Colorado Anschutz Medical Campus, Aurora, Colorado, United States of America, 4 Department of Cancer Immunology and Virology, Dana-Farber Cancer Institute, Boston, Massachusetts, United States of America, 5 Department of Medicine, Harvard Medical School, Boston, Massachusetts, United States of America, 6 Center for Drug Discovery, Children's Healthcare of Atlanta, Atlanta, Georgia, United States of America

☯ These authors contributed equally to this work.
* mamuka.kvaratskhelia@cuanschutz.edu (MK); kyungjin.kim@stpharm.co.kr (KK); baek.kim@emory.edu (BK)

**Data Availability Statement:** The authors declare that all data required to support the findings of this study are available and have been reported within

## Abstract

Allosteric integrase inhibitors (ALLINIs) are a class of experimental anti-HIV agents that target the noncatalytic sites of the viral integrase (IN) and interfere with the IN-viral RNA interaction during viral maturation. Here, we report a highly potent and safe pyrrolopyridine-based ALLINI, STP0404, displaying picomolar $IC_{50}$ in human PBMCs with a >24,000 therapeutic index against HIV-1. X-ray structural and biochemical analyses revealed that STP0404 binds to the host LEDGF/p75 protein binding pocket of the IN dimer, which induces aberrant IN oligomerization and blocks the IN-RNA interaction. Consequently, STP0404 inhibits proper localization of HIV-1 RNA genomes in viral particles during viral maturation. Y99H and A128T mutations at the LEDGF/p75 binding pocket render resistance to STP0404. Extensive *in vivo* pharmacological and toxicity investigations demonstrate that STP0404 harbors outstanding therapeutic and safety properties. Overall, STP0404 is a potent and first-in-class ALLINI that targets LEDGF/p75 binding site and has advanced to a human trial.

## Author summary

Allosteric integrase inhibitors (ALLINIs) target the interaction site of HIV-1 integrase (IN) with host LEDGF/p75 protein, which results in inhibition of IN binding to viral RNA genomes, hence interfering with viral maturation. Here, we report a highly potential and safe ALLINI, STP0404, that displays picomolar to single-digit nanomolar $IC_{50}$ values against different HIV-1 strains. *In vitro* and *in vivo* PK and toxicity studies also show that

this paper and Supplementary Information. Raw data are available (1) WuXi AppTec: https://www.wuxiapptec.com/contact Tel: +86 (21) 2066-3734 Email: wuxiconcierge@wuxiapptec.com (2) Southern Research: https://southernresearch.org/ Infectious Disease Research Facility Tel: 301-694-3232 / 30-694-7223 (3) INA Research Inc.: www.ina-research.co.jp Phone: +81, 2657 26616.

**Funding:** This work was funded by NIH (AI141327) to BK;: Research contract, ST Pharm, Ltd (www.stpharm.co.kr) to BK; NIH (AI143649) to MK and (AI039394) to ANE. The funders had no role in study design, data collection and analysis, decision to publish, or preparation of the manuscript.

**Competing interests:** I have read the journal's policy and the authors of this manuscript have the following competing interests: S.A., U.K., and K.K. are employees of ST Pharm, B.K. is consultant of ST Pharm, and A.N.E. consults for ViiV Healthcare, Co.. All other authors declare no competing interests.

STP0404 exhibits PK profiles optimal for oral once-daily administration. The outstanding efficacy and safety profile of STP0404 have laid the foundation to advance STP0404 into human trials, and STP0404 became the first-in-class ALLINI under clinical trial, which targets the host LEDGF/p75 protein interaction site of HIV-1 IN.

## Introduction

HIV-1 integrase (IN) catalyzes the integration of viral DNA into host chromosomes, and IN is one of the major anti-viral targets [1, 2]. All clinically available HIV-1 IN inhibitors, such as Raltegravir (Ral) and Elvitegravir, target the catalytic site of IN that requires metal ions for its enzymatic reaction, and mainly block the strand transfer activity of IN (IN strand transfer inhibitors, INSTIs) [3–6]. While, together with reverse transcriptase (RT) inhibitors, INSTIs are key components of current anti-retroviral therapy (ART), concerns about their toxicity and resistance demand new and diverse classes of agents with novel anti-viral mechanisms, unique physiochemical characteristics, and desirable safety profiles.

During viral integration, HIV-1 hijacks a host transcription regulator protein, LEDGF/p75, which preferentially directs integration into active transcription units [7–11]. Small molecule inhibitors that target the V-shaped pocket at the IN catalytic core domain (CCD) dimer interface where LEDGF/p75 binds have been designed [12, 13]. Mechanistic studies have elucidated that the primary mode of action of these and related compounds, which are collectively referred to here as allosteric IN inhibitors (ALLINIs; also known as non-catalytic site integrase inhibitors (NCINIs), LEDGINs or INLAIs), is through inhibiting virion maturation [14–20]. Specifically, ALLINIs induce aberrant IN multimerization and interfere with its binding to the viral RNA genome [14]. As a result, viral ribonulceoprotein complexes are mislocalized outside of the protective capsid shell in eccentric virions produced in the presence of ALLINIs [14–20]. While numerous attempts to discover and develop ALLINIs with various chemical scaffolds such as quinoline, benzothiazole, indole and pyridine were made [13, 21–27], none of these candidates has been successfully moved to human clinical trials. Clinical advancement of previously reported highly potent derivatives such as GS-9822 was primarily impeded by compound toxicity observed preclinically in animals [27]. Here, we report a highly potent and safe ALLNI platform with a unique pyrrolopyridine-based scaffold, STP0404. The high antiviral potency, absence of animal toxicity, and oral once-daily pharmacological profiles of STP0404 laid the foundation for advancing STP0404 into phase I clinical trials.

## Results

### Anti-viral activity and therapeutic evaluations in tissue culture systems

Multiple chemical scaffolds such as quinoline (BI224436, **Fig 1A**), benzothiazole (GS-9822, **Fig 1B**) and pyridine (KF116, **Fig 1C**) were previously explored for the discovery of ALLINIs [21, 23, 24, 27, 28]. Here, we employed the pyrrolopyridine-based scaffold to synthesize an ALLINI candidate compound, STP0404 (**Fig 1D**). STP0404 harbors the tert-butoxy moiety that is common to other ALLINIs (**Fig 1**). First, we measured its *in vitro* anti-HIV-1 efficacy and cytotoxicity using various HIV-1 strains in multiple cell types, with cytotoxicity initially tested up to 10 μM (**Table 1**). STP0404 displayed $IC_{50}$ (concentration inhibiting 50% viral production) of 0.41 nM against HIV-$1_{NL4-3}$ without observable cytotoxicity in human PBMCs at 10 μM ($TC_{50}$ >10μM). The calculated *in vitro* minimum therapeutic index (TI) of STP0404 under these conditions was accordingly >25,000. For comparison, we employed two FDA-approved HIV-

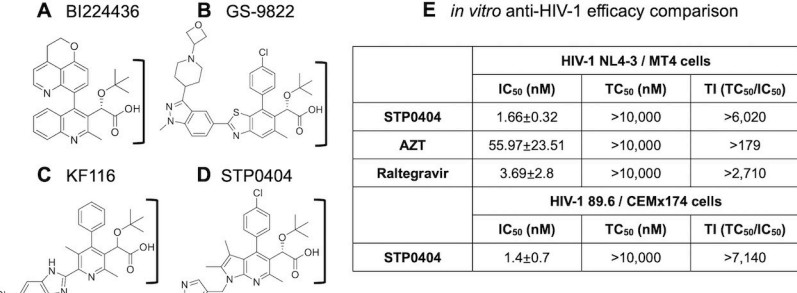

**Fig 1. ALLINI structures and STP0404 activities. A.** Quinoline-based BI224436, **B.** Benzothiazole-based GS-9822, **C.** Pyridine-based KF116, **D.** Pyrrolopyridine-based STP0404. The conserved chemical moiety containing tert-butoxy and carboxylic acid groups are marked by "]". **E.** *In vitro* anti-HIV-1 efficacy of STP0404 against HIV-1$_{NL4-3}$ and 89.6 strains in indicated cell lines. **TI:** Minimal therapeutic index values were calculated with 50% *in vitro* inhibitory concentration values ($IC_{50}$) and 10 μM where no sign of cytotoxicity was observed during tissue culture cytotoxicity measurements ($TC_{50}$) of all tested compounds in both MT4 and CEMx174 cell lines.

1 inhibitors, AZT (RT inhibitor) and Raltegravir (Ral, INSTI), alongside 5 different Ral-resistant clinical isolates (**Table 1**). STP0404 also exhibited picomolar $IC_{50}$ ranges against these resistance strains in PBMCs, with cumulative TI values >125,000. Next, we measured $IC_{50}$, $TC_{50}$ and TI values of STP0404, AZT, and Ral in MT4 cells, where STP0404 displayed similar $IC_{50}$ and minimal TI values as Ral (**Fig 1E**). Finally, STP0404 inhibited dual tropic HIV-1$_{89.6}$ at 1.4 nM $IC_{50}$ in CEMx174 cells (**Fig 1E**). Overall, these tissue culture data demonstrate that STP0404 is a highly potent ALLINI with picomolar to single-digit nanomolar $IC_{50}$ values that inhibits both wild type and Ral-resistant HIV-1 strains.

## Selection and characterization of STP0404 resistant HIV-1 mutants

Next, we selected for STP0404 resistance via serial passage of HIV-1$_{89.6}$ in CEMx174 cells. As shown in **Fig 2**, we conducted two independent passage experiments (blue and grey lines) via gradual elevation (red line) of STP0404 concentration, and monitored HIV-1 in the cell supernatant via p24 level (blue and grey lines). Viral passage was initiated at $IC_{90}$ of STP0404 (21 μM, grey dotted line), and each passage was cultured for 5 days with 1/10 dilution of virus from the previous passage. In both experiments, we observed the appearance of three major spikes of p24 levels at passages 2–4, 10 and 14, indicating the selection of STP0404 resistant viruses at these passages.

**Table 1. *In vitro* anti-viral efficacy of STP0404 against HIV-1$_{NL4-3}$ and Raltegravir-resistant clinical isolates in human PBMCs.** Minimal therapeutic index values (**TI**) were calculated with 50% *in vitro* inhibitory concentration values ($IC_{50}$) and 10 μM where no sign of cytotoxicity was observed during tissue culture cytotoxicity measurements ($TC_{50}$) of all tested compounds in human PBMCs. Data are presented as means of triplicates with standard deviations from the means.

| | NL4-3 | | | Ral-resistant (4736_2) | | | Ral-resistant (4736_4) | | |
|---|---|---|---|---|---|---|---|---|---|
| | IC50 (nM) | TC50 (nM) | TI (TC50 / IC50) | IC50 (nM) | TC50 (nM) | TI (TC50 / IC50) | IC50 (nM) | TC50 (nM) | TI (TC50 / IC50) |
| STP0404 | 0.41±0.43 | >10,000 | >24,000 | 0.03±0.04 | >10,000 | >330,000 | 0.04±0.06 | >10,000 | >250,000 |
| AZT | 8.92±2.08 | >10,000 | >1,121 | 0.42±0.37 | >10,000 | >23,000 | 6.71±5.74 | >10,000 | >1.490 |
| Raltegravir | 0.44±0.74 | >10,000 | >22,700 | >50 | >10,000 | ~200 | 36.48±53.6 | >10,000 | >270 |
| | Ral-resistant (8070_1) | | | Ral-resistant (8070_2) | | | Ral-resistant (1556_1) | | |
| | IC50 (nM) | TC50 (nM) | TI (TC50 / IC50) | IC50 (nM) | TC50 (nM) | TI (TC50 / IC50) | IC50 (nM) | TC50 (nM) | TI (TC50 / IC50) |
| STP0404 | 0.06±0.07 | >10,000 | >166,000 | 0.23±0.39 | >10,000 | >43,400 | 0.08±0.04 | >10,000 | >125,000 |
| AZT | 6.71±5.74 | >10,000 | >1,490 | 79.9±11.76 | >10,000 | >125 | 9.33±0.22 | >10,000 | >1,000 |
| Raltegravir | >1,000 | >10,000 | ~10 | >100 | >10,000 | ~100 | >500 | >10,000 | ~20 |

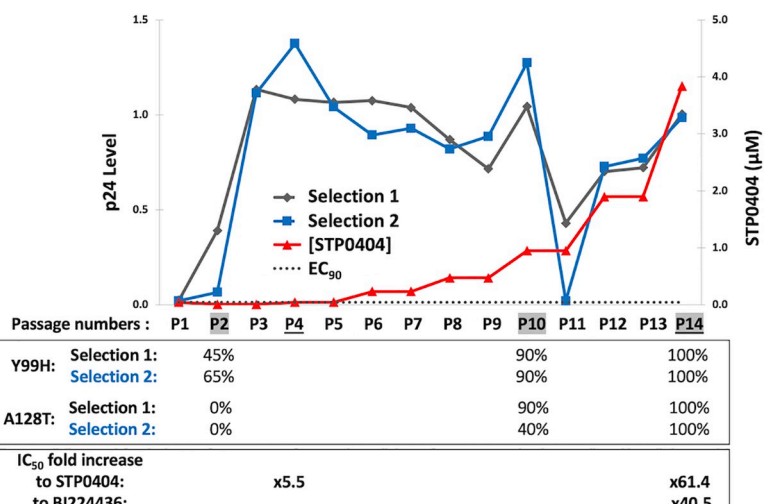

**Fig 2. Selection of STP0404-resistant HIV-1 variants.** Two independent *in vitro* serial passages (grey and blue lines) of HIV-1$_{89.6}$ in CEMx174 cells were initiated at STP0404 IC$_{90}$ concentration (black dotted line). Viral production at each passage was monitored by p24 ELISA of collected media, and STP0404 concentration (red line) was elevated when p24 level spikes were observed. 20 cloned IN sequences were determined for three viral populations (passages, 2, 10, and 14) in both viral selections, with percentages of resistance Y99H and A128T mutation populations noted. Fold increases in IC$_{50}$ values of the viral populations collected at passage 4 against STP0404 and passage 14 against STP0404 and BI224436 are indicated.

Next, we characterized the viruses collected at passages 2, 10 and 14. First, we cloned and sequenced the viral IN regions (20 IN clones per passage). As shown in **Fig 2**, Y99H was initially selected at comparatively low STP0404 concentration at passage 2, and was detected in 90% of the sequenced IN clones at passage 10. At passage 10, the A128T change was partially detected (90 and 40%) in both selection cultures together with Y99H. However, when STP0404 concentration were further elevated, all 40 sequenced IN clones in both cultures contained both Y99H and A128T (passage 14). Next, we determined STP0404 IC$_{50}$ values against the viruses collected at passage 4 and 14. As shown in **Fig 2**, the viral population collected at passage 14, which contained both Y99H and A128T changes, was 61.4-fold less susceptible compared to wild type HIV-1$_{89.6}$, while the passage 4 viruses were about 5.5 times less susceptible. For comparison, the passage 14 virus population exhibited 40.5-fold reduced sensitivity to quinoline-based BI224436 compared to wild type HIV-1$_{89.6}$, indicating that the mutations selected against STP0404 exhibit cross-resistance with respect to other previously developed ALLINIs.

## Replication kinetics of STP0404-resistant HIV-1 variants and STP0404 effect with polymorphic IN changes

Next, we tested replication kinetics of the STP0404 resistant viruses in CEMx174 cells. For this, we constructed molecular clones of HIV-1$_{89.6}$ containing Y99H alone, A128T alone, and both Y99H and A128T IN changes. Equal levels of these viruses were used to infect CEMx174 cells, and p24 levels in the media were determined daily for 4 days. As shown in **Fig 3A**, while the single A128T mutant replicated like wild type, Y99H and Y99H/A128T mutant viruses yielded 35 and 70-fold less p24, respectively, at day 4 post-infection, suggesting that Y99H significantly impaired HIV-1 replication kinetics.

Certain IN positions in and around the ALLINI binding pocket are highly polymorphic among circulating HIV-1 strains, including 124 (A124, T124, N124) and 125 (T125, and A125)

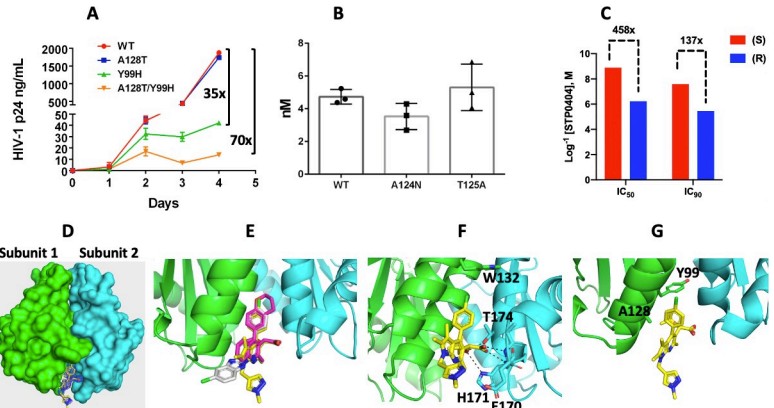

**Fig 3. Characterization of STP0404 resistant viral replication kinetics, efficacy against common polymorphisms, antiviral efficacy of enantiomers, and X-ray structure of IN CCD–STP0404 complex. A.** Replication kinetics of STP0404 resistant HIV-1 variants in CEMx174 cells. p24 levels of wild type (WT) HIV-1$_{89.6}$ or indicated IN variant in the collected culture media are noted. Fold differences of the variants in p24 levels, compared to wild type virus culture at day 4 were calculated. **B.** IC$_{50}$ values of STP0404 against wild type HIV-1$_{89.6}$ or indicated IN variants in CEMx174 cells. **C.** IC$_{50}$ values of (R) and (S) enantiomers of STP0404 against wild type HIV-1$_{89.6}$ in CEMx174 cells were determined, and their fold differences were calculated. Data in panels a-c are presented as means of triplicates and error bars indicate the standard deviations from the means. **D.** X-ray structure of HIV-1 IN CCD dimer bound to STP0404 (yellow/mesh). Two interacting CCDs are colored green (subunit 1) and cyan (subunit 2). Fo-Fc omit map calculated while omitting STP0404 is contoured at 3σ at resolution of 2.2 Å. A close up of STP0404 (mesh) binding to the LEDGF/p75 binding pocket was shown in **S2 Fig**. **E.** Superposition of our STP0404 structure (yellow, PDB: 7KE0) with quinoline based BI-D (pink, PDB: 4ID1) and pyridine based KF116 (gray, PDB: 4O55). **F.** Principal STP0404 interactions with the CCD-CCD dimer. W132, E170, H171 and T174 side chains are shown as sticks. Dashed lines indicate hydrogen bonding contacts. **G.** The side chains of A128 and Y99 are shown in the context of bound STP0404. Data collection and refinement statistics are described in **S1 Table**.

[26, 29]. HIV-1$_{NL4-3}$ and 89.6 strains that were used in this study (**Fig 1E** and **Table 1**) harbor T124 and A124, respective, as well as T125. To ascertain susceptibility to other common amino acids, we evaluated STP0404 efficacy against HIV-1$_{89.6}$ carrying N124 or A125 (**Fig 3B**). STP0404 retained similar low IC$_{50}$ values against these variant viruses, suggesting STP0404 can effectively control natural HIV-1 variants that contain different amino acid sequences near the region that confers STP0404 resistance.

## Anti-HIV-1 efficacy of STP0404 enantiomers

Due to chirality at the tert-butoxy moiety, STP0404 can exist as enantiomers (**Fig 1D**), although we note that the racemic mixture of STP0404 contained >95% of the (S) form. To assess enantiomer-specific activities, we compared the IC$_{50}$ values of purified STP0404 (S) and (R) forms. Based on IC$_{50}$ and IC$_{90}$ values, the (R) form was ~460- and ~140-less potent, respectively, than the (S) form (**Fig 3C**), and both the racemic mixture used in **Fig 1E** and (S) form displayed very similar anti-HIV-1 potencies (1.4 and 1.3 nM, respectively). These data support that the (S) form of STP0404, which predominantly exists in the racemic mixture, underlies the anti-HIV efficacy of STP0404.

## X-ray structure of HIV-1 IN CCD–STP0404 complex

Next, we determined the X-ray structure of the HIV-1 IN catalytic core domain (CCD) (F185H) in complex with STP0404 at 2.2 Å resolution (**S1 Table**). As shown in **Fig 3D**, STP0404 binds to the V-shaped pocket formed by two interacting CCD subunits, which is also the principal binding site for LEDGF/p75 [12]. Superposition of the STP0404 structure with prior quinoline based BI-D and pyridine based KF-116 bound structures [20, 23] revealed that

STP0404 maintains many of the critical interactions of the ALLINI family (**Fig 3E**). The carboxylic acid moiety of STP0404 is engaged in a bi-dentate hydrogen bonding interaction with the backbone amides of E170 and H171 of subunit 2 of the CCD dimer with an additional hydrogen bonding interaction with the protonated Nδ of H171 (**Fig 3F**). The hydroxyl group of T174 and the protonated Nδ of H171 also interact with the ether oxygen of the tert-butoxy moiety of STP0404. The chloro-phenyl group extends up into the hydrophobic pocket, which is capped by W132 of subunit 1. The pyrazole moiety of the compound extends out from the core of the compound and does not directly interact with the CCD dimer. However, there could be important interactions between this group and the C-terminal domain (CTD) in the context of a full-length IN as seen with other ALLINIs [30]. While A128 lies in close proximity to the pyrrolopyridine ring of STP0404, its substitution with bulkier and polar Thr is likely to create steric hindrance for the inhibitor (**Fig 3G**). Y99 is positioned deep inside the V-shaped pocket and contributes to CCD-CCD interactions. Accordingly, the Y99H mutation may alter pocket shape to indirectly modulate IN-STP0404 binding. This putative Y99H-based structural change would appear to underlie viral resistance to STP0404 at comparatively low inhibitor concentrations, whereas higher levels of resistance require both A128T and Y99H changes.

## Effect of STP0404 on IN-RNA interaction and HIV-1 particle morphology

A series of reports revealed that the key mode of ALLINI action is to interfere with HIV-1 maturation [14–20]. More specifically, ALLINI-induced aberrant oligomerization/aggregation inhibits IN interaction with viral RNA genomes in viral particles [14]. Viruses produced from ALLINI-treated cells accordingly show a distinct morphological defect, with their RNA genomes mislocalized outside of viral capsids [14, 20]. First, we used an Alpha Screen assay to test whether STP0404 interferes with IN-RNA binding *in vitro*. This assay biochemically determined the effect of STP0404 on the interaction between tagged IN protein and HIV-1 TAR RNA. As shown in **Fig 4A**, STP0404 inhibited ($IC_{50}$ = 0.020 ± 0.003 μM) IN-RNA binding.

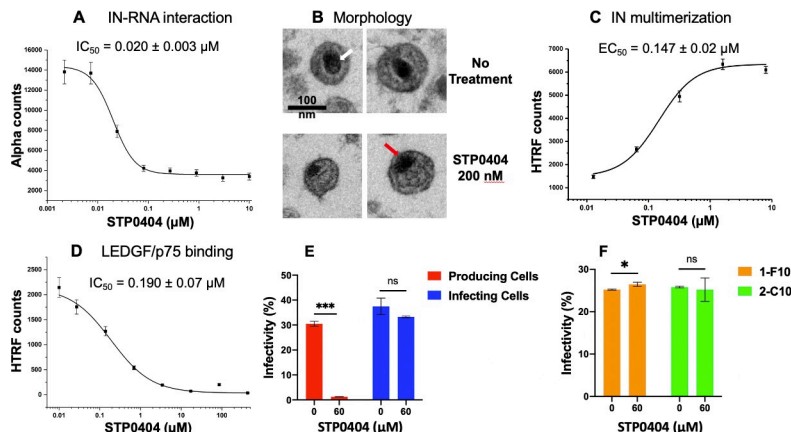

**Fig 4. Effects of STP0404 on IN-RNA binding, HIV-1 particle morphology, IN multimerization, LEDGF/p75 binding, and early versus late stages of HIV-1 replication. A.** Inhibition of IN-RNA binding. **B.** TEM images of HIV-1 particles. **C.** Effect of STP0404 on IN multimerization. **D.** Effect of IN binding to LEDGF/p75. Data in panels **A**, **C** and **D** are means of triplicates with error bars indicating standard deviations from the means. **E.** STP0404 effect on early versus late stages of HIV-1 replication. Jurkat cells were transduced with HIV-1 GFP vector produced from 239T cells treated with and without STP0404 (0 and 60 nM), indicated as *"producing cells"*. Jurkat cells pre-treated with STP0404 (0 and 60 nM) were transduced with HIV-1 GFP vector produced from untreated 293T cells, indicated as *"infecting cells"*. The transduction efficiency was determined by FACS for GFP expression. **F.** LEDGF/p75 effect on STP0404 efficacy. Two independent LEDGF/p75 knockout Jurkat cell lines (1-F10 and 2-C10; cite PMID 32994325) were pretreated with STP0404 (0 and 60 nM) and transduced with HIV-GFP. Transduction efficiency was determined by FACS. Data in panels e and f are presented as means of triplicates and error bars indicate the standard deviations from the means. *P*-value < 0.05 is represented as *; *p-value* < 0.001 is represented as ***; *ns* indicates not significant.

Next, we examined the morphology of HIV-1$_{89.6}$ particles produced from 293 T cells transfected with HIV-1 89.6 plasmid with and without STP0404 treatment. Typically, in transmission electron microscopy (TEM), the viral RNA genomes, which are tightly packed with HIV-1 nucleocapsid protein inside the viral core, are identified via their electron density (see arrows in **Fig 4B**). While viruses produced in the absence of the inhibitor as expected revealed viral RNA genomes inside the viral capsid (white arrow), viruses produced from STP0404-treated cells showed their viral RNA genomes outside of the capsid (**Fig 4B**, red arrow). These data support that STP0404, which inhibits IN-RNA binding, results in mislocalized viral RNA in produced virus particles.

## Effect of STP0404 on IN multimerization and LEDGF/p75 binding

A key mode of action of ALLINIs is to induce high-order aberrant IN multimerization, which consequently interferes with IN binding to RNA [14, 30, 31]. To investigate the ability of STP0404 to induce higher-order IN multimerization, we employed a homogenous time resolved fluorescence (HTRF)-based assay [32]. This assay biochemically determined the effect of STP0404 on the proximity/multimerization between two full-length IN protein populations differentially labeled, and HTRF signal was used to determine EC$_{50}$ values. **Fig 4C** shows that STP0404 induced higher-order IN multimerization at EC$_{50}$ value $0.147 \pm 0.02$ μM. Next, we examined the ability of STP0404 to inhibit IN binding to LEDGF/p75 using another HTRF-based assay [33], which monitors the direct binding of IN protein to LEDGF/p75 protein, which both are differentially tagged. As shown in **Fig 4D**, STP0404 inhibited IN-LEDGF/p75 binding with IC$_{50}$ = $0.190 \pm 0.07$ μM. Overall, these data demonstrate that STP0404 inhibits IN-RNA binding by inducing aberrant IN multimerization, and can also interfere with IN binding to LEDGF/p75.

## Anti-HIV effect of STP0404 in producing vs. infecting cells

Since viral maturation occurs as the last (late) step of the HIV-1 life cycle, we tested the effects of STP0404 on the early vs late steps of viral replication using a single-round HIV-1 GFP vector and Jurkat cell lines (**Fig 4E**). HIV-1 produced from 293T cells (*producing cells*) treated with STP0404 (60 nM) showed significantly reduced transduction efficiency as compared to virus produced from untreated 293T cells. However, HIV-1 vector produced from untreated 293T cells showed no reduction in transduction efficiency (*infecting cells*) when Jurkat cells were treated with this same STP0404 concentration (**Fig 4E**), suggesting that STP0404 does not or only weakly inhibits the early steps of HIV-1 replication including integration. Addition of 60 nM STP0404 to LEDGF/p75 knockout Jurkat T cells also failed to significantly inhibit infection (**Fig 4F**), revealing that STP0404 does not inhibit the early steps of HIV-1 replication regardless of LEDGF/p75 expression at drug concentrations that fully inhibited the late-stage maturation step.

## *In vitro* and *in vivo* drug metabolism and pharmacokinetics (DMPK) of STP0404

As STP0404 displayed highly potent anti-HIV-1 activities *in vitro*, we evaluated DMPK to assess its safety and efficacy as a prelude to human clinical trials. In *in vitro* absorption, distribution, metabolism, and excretion (ADME) studies (**S2 Table**), STP0404 displayed extended microsomal stability ($>133.3$ min T$_{1/2}$) and plasma stability (close to 100%) in all tested species. Interestingly, STP0404 showed much longer metabolic stability in hepatocytes in both monkey and human settings. *In vitro* protein binding of STP0404 in rat, dog, monkey and human plasma (93.9–99.7%) was similar across species. STP0404 revealed low (high μM) or no

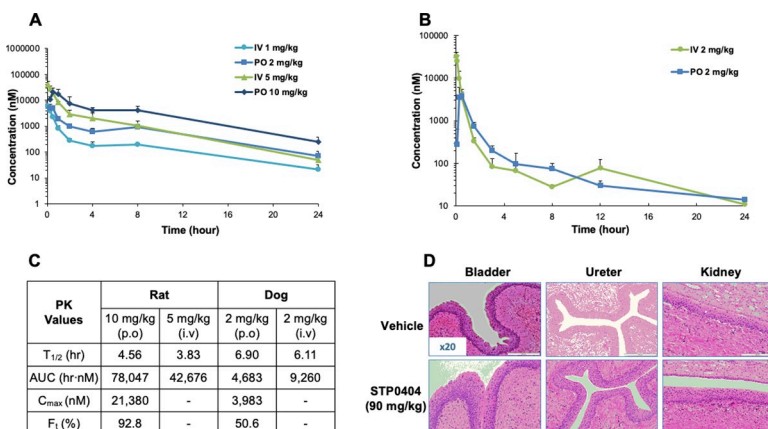

**Fig 5. _In vivo_ pharmacokinetics and histology of STP0404 treated subjects. A.** SD rats were administrated with STP0404 by IV (1 and 5 mg/kg) and PO (2 and 10 mg/kg). **B.** Beagle dogs were administrated IV and PO (2 mg/kg). Plasma concentrations were determined by LC-MS for 24 h post administration (see Methods). For **A** and **B**, data are presented as means of three independent experiments and error bars indicate the standard deviations from the means. **C.** Half-lives ($T_{1/2}$), area under the curve (AUC), maximum concentration ($C_{max}$), and bioavailability ($F_t$) from the administrated animals (**A** and **B**) were calculated. **D.** Histology of beagle dogs orally administrated with STP0404 (90 mg/kg) or vehicle for bladder, ureter, and kidney in 4-week GLP toxicology study. The entire GLP toxicology study's results were summarized in **S3 Table**.

inhibition (NI) of 8 different cytochrome P450 enzymes, while STP0404 required comparatively high concentrations ($EC_{50}$) to activate drug metabolizing factors (CYP3A and PXR).

We next investigated _in vivo_ PK studies of STP0404 in male SD rats (**Fig 5A**) and beagle dogs (**Fig 5B**) following intravenous bolus (1 and 5 mg/kg) or oral administration (2 and 10 mg/kg) regimens. As summarized in **Fig 5C**, the half-life ($T_{1/2}$) was 3–7 h, and oral bioavailability ($F_t$) was 50–93% in these two animal species. Also, systemic exposure, which was determined by area under the curve and maximum concentration of STP0404 in plasma (AUC and $C_{max}$), increased dose dependently from 2 to 10 mg/kg. Based on the half-life and bioavailability data in dogs and rats, we concluded that STP0404 displays appropriate PK profiles for once daily administration.

## Toxicology of STP0404

As summarized in **S3 Table**, STP0404 lacked both gene mutation-inducing potential across five bacterial strains and chromosomal aberration-inducing potential in CHL/IU cells (25~140 μg/mL). Furthermore, STP0404 lacked micronucleus-inducing and bone marrow cell proliferation inhibitory potentials in rats (500, 1000 and 2000 mg/kg/day), supporting that STP0404 is not genotoxic.

Finally, repeated dose toxicology studies were used to obtain Maximum Tolerability Dose (MTD) in good laboratory practice (GLP) system in rats and dogs (**S3 Table**). In a four-week oral dose toxicology study of STP0404 in rats, body weight decreased >10% only in high-dosed males (600 mg/kg/day), and no-observed-adverse-effect level (NOAEL) of STP0404 was thus determined at 300 mg/kg/day for males and 600 mg/kg/day for females. In the four-week oral dose toxicology study in dogs, no abnormal changes were noted at any dose level (30 mg/kg, 60 mg/kg and 90 mg/kg). Therefore, the NOEAL of STP0404 was determined to be 90 mg/kg/day for both male and female dogs. By histopathology, in contrast to the most advanced benzothiazole-based ALLINIs [27] lesions were not observed in any organ, including intestine, bladder, ureter and kidney (**Fig 5D**). Also, STP0404 did not induce any safety pharmacology concerns in central nervous system in rats, respiratory system in rats or cardiovascular system

in dogs (**S3 Table**). Finally, the highest non-severely toxic dose (HNSTD) was defined based on the high-dose level (90 mg/kg/day) in the four-week dog toxicology study in accordance with International Conference on Harmonisation (ICH) guidance S9. The first human dose, calculated by human equivalent dose (HED) with safety factor 10, was determined to be 340 mg based on a 70 kg human body weight. The first in human (FIH) administration of STP0404, starting at 200 mg once a day by oral administration with safety factor 15, is on-going. Overall, our pharmacological and toxicological evaluations generated information essential for the human trial of STP0404.

## Discussion

We have discovered STP0404, a pyrrolopyridine-based ALLINI with a highly potent activity against multiple HIV-1 strains in multiple cell types with excellent therapeutic index values. Our structural, biochemical and virological studies have validated that STP0404 is an ALLINI that binds to the LEDGF/p75 binding site of IN dimers and inhibits viral maturation by inter-fering with the IN-RNA interaction and mislocalizing viral RNA genomes in the produced viral particle.

The Y99H mutation has been much less commonly reported compared to other known ALLINI resistance mutations such as A128T [13, 34]. Y99 lies deep inside the V-shaped pocket, and our X-ray crystal structure of the IN-STP0404 complex failed to reveal a direct contact between STP0404 and Y99 (**Fig 3**). Possibly, the Y99H mutation may induce a confor-mational change near the STP0404 binding site that assists the molecular clash between the substituted T128 residue and STP0404. Note that in two independent STP0404 resistance selection experiments (**Fig 2**), Y99H was selected first at comparatively low STP0404 concen-tration, while A128T was subsequently acquired to confer full resistance against STP0404. Therefore, the putative conformational change made by Y99H appears to be sufficient to block compound binding at low STP0404 concentrations. It is interesting, however, that while the A128T IN mutant virus replicated similarly to the wild type, the Y99H IN mutant as well as Y99H/A128T virus displayed severely defective replication capability, implying the unfit nature of STP0404 resistance viruses. Interestingly, when we continued the culture of the cloned Y99H mutant virus for additional three passages in the presence and absence of STP0404 (12 nM), we still did not observe any additional mutations in the cloned IN genes. This suggests the Y99H mutant did not improve its unfit phenotype by gaining any compensa-tory IN mutations during these three passages regardless of the STP0404 treatment. We also conducted *in vitro* efficacy test of STP0404 against SIVmac239. Unlike Raltegravir, STP0404 failed to inhibit SIVmac239 (S1 **Fig**), and BI224436 ALLINI also failed to inhibit SIVmac239. The failure of these ALLINIs to inhibit SIVmac239 likely results from the significant sequence variations between HIV-1 and SIVmac239 (e.g. L99 and M128) at the V-shape LEDGF/p75 binding site of these lentivirus IN proteins, which does not allow the animal efficacy test of ALLINIs using SIV systems. Also, these ALLINIs are likely incapable of inhibiting HIV-2 because HIV-2 (e.g. HIV-2 Rod) IN protein also encode L99 and M128.

Our preclinical investigations indicate that STP0404 is a safe compound. Our PK studies demonstrate that STP0404 is rapidly absorbed, with high to intermediated oral bioavailability in rats and dogs (92.8% and 50.6%, respectively). The four-week repeated toxicity study in bea-gle dogs with oral administrations **S3 Table** support that the single dose of STP0404 was well tolerated up to 500 mg/kg. Repeated oral dosing of STP0404 was well tolerated in beagle dogs, and, based on the 4-week once daily oral dose toxicology study in dogs, NOAEL of STP0404 was determined to 90 mg/kg. It is difficult to directly compare the PK profiles of STP0404 with other previously reported ALLINIs (BI224436 and GS9822) because different animal species

and strains were used [27, 35]. However, the PK data of STP0404 clearly supports its oral once-daily administration route. Indeed, based on these encouraging preclinical findings, we have recently started phase I clinical studies with 200 mg (1/15) STP0404 in a Single Ascending Dose (SAD) regimen. Importantly, since there is no efficacy animal model for HIV-1, we have applied toxicity standards for the dose determination without considering efficacy. Indeed, we decided that the FIH dose would be <340 mg/70 kg adults, which is one tenth of the value calculated by human capacity [36].

Overall, our extensive mechanistic investigations show that STP0404 is a potent pyrrolo-pyridine-based ALLINI that inhibits HIV-1 maturation. The broad *in vivo* preclinical PK and toxicity investigations allowed us to determine the human dose and application method of STP0404. Collectively, our studies have laid the foundation to advance STP0404 into human trials, and STP0404 became the first-in-class ALLINI under clinical trial, which targets the host LEDGF/p75 protein interaction site of HIV-1 IN.

## Methods

### Ethics statement

All protocols involving animal experimentation were reviewed and approved by the respective Animal Care and Use Committee of each test facility. Study numbers for respective experiments were described in subsequent method sections of each assay. PK studies were performed at WuXiAppTec (Shanghai, China) Co., Ltd and adhered to the study protocol and Standard Operating Procedures (SOPs), but was not intended to be in full compliance with international good laboratory practice (GLP) regulations. All chemicals used were reagent grade or better.

### Virus culture

The cell lines (CEMx174 and Jurkat cells) were cultured at 37°C in 5% $CO_2$ in RPMI 1640 medium with L-glutamine (Corning) supplemented with 10% fetal bovine serum (Omega Scientific) and penicillin-streptomycin. Cell-free supernatants were measured for p24 or p27 capsid antigen content utilizing a commercial p24 or p27 ELISA kit (Advanced Bioscience Laboratories).

### *In vitro* anti-viral assay in cell lines

CEMx174 cells were infected with HIV-1 primary isolate 89.6 (From infectious clone p89.6, NIH AIDS Reagent Program) at approximately at 10% of cell population determined by FACS analyses. To determine antiviral activity during virus production, STP0404 was added at concentration in the range of 0.1 nM—10 μM during media exchange at 4 hrs post-infection. DMSO was used as a negative control. Cell-free supernatants were measured for p24 antigen production, 5 days post-infection. The anti-HIV-1 efficacy of the two enantiomers of STP0404 were also determined by using the same protocol. All assays were conducted in triplicates. The $IC_{50}$ values were computed using GraphPad Prism (Version 9) and presented as means ± S.D. of the triplicates. For SIVmac239, CEMx174 cells were infected with viral supernatant containing SIVmac293 at 50,000 TCID50/mL (kind gift from Dr. G. Silvestri, Emory Yerkes National Primate Research Center) with 100 μL/$10^6$ cells at varying concentrations of STP0404, BI224436, and Raltegravir, and the same above protocol was followed except for that p27 capsid antigen content was measured daily for 5 days post infection.

### *In vitro* anti-viral assay in human PBMCs

For the assay with human PBMCs, PHA stimulated cells from at least two pooled healthy were infected with HIV-1 strains in 96-well plates. Test drug dilutions, which were prepared at a 2X

concentration in microtiter tubes and 100 μL of each concentration (nine total concentrations), were placed in appropriate wells using the standard format. 50 μL of a predetermined dilution of virus stock was placed in each test well (final MOI $\cong$ 0.1). The PBMC cultures were maintained for 7 days following infection at 37 $^{\circ}$C, 5% $CO_2$. After this period, cell-free supernatant samples were collected for analysis of reverse transcriptase (RT) activity. All assays were conducted in triplicates. For RT assay, a microtiter plate-based reverse transcriptase (RT) reaction was utilized (Buckheit et al., AIDS Research and Human Retroviruses 7:295–302, 1991). Tritiated thymidine triphosphate ($^3$H-TTP, 80 Ci/mmol, NEN) is received in 1:1 $dH_2O$: Ethanol at 1 mCi/mL. Poly rA:oligo dT template:primer (GE Healthcare) was prepared as a stock solution by combining 150 μL poly rA (20 mg/mL) with 0.5 mL oligo dT (20 units/mL) and 5.35 mL sterile $dH_2O$ followed by aliquoting (1.0 mL) and storage at—20˚C. The RT reaction buffer was prepared fresh on a daily basis and consisted of 125 μL 1.0 M EGTA, 125 μL $dH_2O$, 125 μL 20% Triton X100, 50 μL 1.0 M Tris (pH 7.4), 50 μL 1.0 M DTT, and 40 μL 1.0 M $MgCl_2$. The final reaction mixture was prepared by combining 1 part $^3$H-TTP, 4 parts $dH_2O$, 2.5 parts poly rA:oligo dT stock and 2.5 parts reaction buffer. Ten microliters of this reaction mixture were placed in a round bottom microtiter plate and 15 μL of virus containing supernatant is added and mixed. The plate was incubated at 37˚C for 60 mins. Following incubation, the reaction volume was spotted onto DE81 filter-mats (Wallac), washed 5 times for 5 mins each in a 5% sodium phosphate buffer or 2X SSC (Life Technologies). Next they were washed 2 times for 1 min each in distilled water, 2 times for 1 min each in 70% ethanol, and then dried. Incorporated radioactivity (counts per min, CPM) was quantified using standard liquid scintillation techniques. These PBMC analysis was conducted by fee to service through Southern Research Institute (Frederick, MD, USA). The $IC_{50}$ were analyzed using GraphPad Prism (Version 9) and presented as means ± S.D. of the triplicates.

## Resistant virus selection

CEM x174 cells infected with HIV primary isolate 89.6 were cultured in the presence of STP0404 at the concentration of $IC_{90}$ determined earlier. At each passage, cells from original culture in the presence of inhibitor ($0.75x10^6$ cells) were mixed with equal amount of no-drug control cells to propagate, and viral replication was monitored by the production of p24 antigen in the supernatant. Inhibitor concentration was gradually increased at each passage when the p24 antigen level was observed to be increased. At passage 2, 10 and 14 ($IC_{90}$ samples), where major spikes of p24 antigen level observed, viral RNA was extracted using QIAexpress (Qiagen) and IN sequences were determined by RT-PCR with DNA primer set targeting HIV-1 IN gene using Superscript III One-Step RT-PCR kit (Invitrogen) followed by ligation into TOPO 2.1 vectors (Invitrogen) for plasmid sequencing. The population of drug-induced mutations at each passage mentioned above was determined by sequencing of at least 10 resulting transformants.

## Characterization of STP0404 resistant and natural IN variant viruses

CEM x174 cells were infected with cell-free viral supernatant collected during virus resistance assays above. Passages 4 and 14 viral supernatant collected were selected to illustrate the effect of single mutation (Y99H, passage 4) and double mutations (Y99H and A128T). The viral stocks were normalized by the amount of p24 capsid antigen, determined to infect CEMx174 cells at approximately 10% of populations (10,000 pg / $10^6$ cells). To determine antiviral activity during virus production, STP0404 was added at concentration in the range of 10 μM—0.1 nM for the passage 4 or 50 μM—10 nM for the passage 14 virus infected cells during media exchange at 4 hrs post-infection. DMSO was used as a negative control. Cell-free supernatants

were measured for p24 antigen production 5 days post-infection. p89.6 plasmid was applied for Quickchange kit (Invitrogen) to create Y99H single, A128T single, Y99H/A128T double mutants as well as two natural variants (A124N and T125A), and these 89.6 variant viruses were generated by transfecting 293T cells and culturing the produced viruses in CEMx174 cells. The IC$_{50}$ of STP0404 against respective viruses were analyzed using GraphPad Prism (Version 9) and presented as means ± S.D. of triplicates. Replication kinetics of the STP0404 resistant variants were compared by infecting CEMx174 cells with an equal p24 level, viral production was determined by p24 assay daily for 5 days.

## Transmission electron microscopy (TEM) analysis

293T cells were transfected with plasmids containing proviral clones encoding WT HIV-1 89.6 using polyethylenimine (PEI). After 6 hrs, cells were rinsed with DPBS gently, and then cells were resuspended in a fresh media with or without the inhibitor (1 μM, final). At 96 hrs post-transfection, cell cultures were centrifuged (1,800 xg) and filtered through 0.45 μm filter to remove cells and any cell debris, and viruses were harvested by centrifugation at 100,000 xg for 1.5 hrs in the presence of TNE sucrose buffer (5% sucrose, final). The pellets were fixed in a buffer containing 0.1 M Na cacodylate at pH 7.4 and 2.5% Glutaraldehyde, and submitted to Emory Integrated Core Facilities for sectioning, followed by TEM. Virus pellets were dehydrated in a graduated ethanol series and embedded in Epon resin. Ultrathin sections were stained using uranyl acetate and observed under a transmission electron microscope (JEOL JEM-1400), equipped with Gatan CCD camera at the Emory Integrated Electron Microscopy Core. Total of approximately 1,000 virions each (with or without the inhibitor treatment) were captured in the images obtained for comparisons in virion morphology.

## HIV-1 IN CCD (F185H) Expression, Purification, Crystallization, and X-ray Crystallography

The HIV-1 IN CCD (residues 50–212) containing the F185H mutation was expressed and purified as described [23]. The protein was concentrated to 8 mg/ml and crystallized using hanging-drop vapor diffusion method with a crystallization buffer consisting of 100 mM sodium cacodylate pH 6.5, 100 mM ammonium sulfate, 10% (w/v) PEG 8000, and 5 mM DTT. Crystallization drops were prepared using an equal volume of protein and well solution. Crystallization trays were prepared on ice at room temperature and then transferred to 4°C for storage. Crystals formed within one week to a month. Crystals were transferred to a drop containing crystallization solution, 5 mM of STP0404, and 10% DMSO. Crystals were soaked overnight prior to data collection. Crystal data were collected on a Rigaku Micromax-007 at 100 K. Data were integrated and scaled using HKL3000 [37] and Scalepack [38]. Phaser [39] in the PHENIX suite [40] was used to run molecular replacement using Protein Data Bank code 4O55 as a search model [23]. Phenix.refine [41] was used for data refinement, and manual refinement was done in Coot [42]. The coordinates are deposited in the Protein Data Bank under accession codes 7KE0. The data and refinement statistics are given in **S1 Table**.

## Inhibition assay for IN binding to RNA

Primary antiviral activity of ALLINIs was observed during virion maturation, where they inhibit IN-RNA interactions [14], we examined the ability of STP0404 to inhibit recombinant IN binding to synthetic TAR RNA using Alpha screen based assay [14]. Briefly, different concentrations of STP0404 were incubated with 100 nM His$_6$ tagged IN in buffer containing 100 mM NaCl, 1 mM MgCl$_2$, 1 mM DTT, 1mg/mL BSA, 25 mM Tris, pH 7.4 at 4°C for 2 hrs. This mixture was then added to the nickel acceptor beads while biotinylated-TAR RNA was added

to the streptavidin donor beads. After 2-hrs incubation at 4˚C, the RNA mixture was added to the IN-drug mixture and the reading was taken after 1 hr incubation at 4˚C by PerkinElmer Life Sciences Enspire multimode plate reader. The $IC_{50}$ values were calculated by OriginLab software.

### IN multimerization assay

To test the ability of STP0404 to induce higher-order multimerization of IN we employed homogeneous time-resolved fluorescence (HTRF)-based assay [33]. Briefly, His-tagged and FLAG-tagged INs (each at 10 nM final concentration) were mixed in buffer containing 25 mM Tris, pH 7.4, 150 mM NaCl, 2 mM $MgCl_2$, 0.1% Nonidet P-40 and 1 mg/ml BSA. This mixture was incubated with various concentrations of the test compounds for 3 hrs at room temperature. The detection is based on anti-His6-XL665 and anti-FLAG-EuCryptate antibodies (Cisbio, Inc., Bedford, MA) which were added to the reaction and incubated at room temperature for 3 hrs. The HTRF signal was recorded by PerkinElmer EnSpire multimode plate reader and OriginLab software was used to calculate the $EC_{50}$ values.

### Inhibition assay for LEDGF/p75 binding to IN

We examined the ability of STP0404 to inhibit IN binding to LEDGF/p75 using another HTRF-based assay [33]. Briefly, 10 nM His-tagged IN was pre-incubated in a binding buffer (150 mm NaCl, 2 mm $MgCl_2$, 0.1% Nonidet P-40, 1 mg/ml BSA, 25 mm Tris, pH 7.4) with the compound for 30 min at room temperature, and then 10 nM FLAG-tagged LEDGF/p75 was added to the reaction. This was followed by addition of 6.6 nM anti-His6-XL665 and 0.45 nM anti-FLAG-EuCryptate antibodies (Cisbio, Inc., Bedford, MA). After overnight incubation at 4˚C, the HTRF signal was recorded and the $IC_{50}$ values were calculated by OriginLab software.

### HIV-1 inhibition in *"producing"* and *"infecting"* cells

To test for STP0404's inhibitory effects during the late stage of HIV-1 life cycle such as the maturation step, we employed the *"producing cells"* setup. In T75 tissue culture flasks, 293T cells were pre-treated with and without STP0404 (0 and 60 nM) for 2 hrs prior to transfection with pD3-HIV (13 μg) and pVSV-g (6 μg), using 0.3 mg/ml polyethylenimine. After 24 hrs incubation at 37˚C, the supernatant was removed and replaced with fresh media with and without STP0404 (0 and 60 nM). 48 hrs later, D3-HIV vector from each treatment group was concentrated from the collected supernatant via ultracentrifugation (22,000 rpm for 2 hrs). Following p24 quantification, using 96-well plates, similar amount of respective D3-HIV vector was used to transduce Jurkat cells in triplicates for 48 hrs. The cells were harvested and fixed with 3.7% formaldehyde before being analyzed for GFP expression using FACS (Miltenyi Biotec, VYB). Separately, the *"infecting cells"* setup was used to investigate the inhibitory effects of STP0404 against the early stages of HIV-1 life cycle such as the integration and transcription steps. D3-HIV vector collected from transfected, non-treated 293T cells were used to transduce LEDGF/p75 +/- Jurkat cells which have been subjected to 2 hrs pre-treatment with and without STP0404 (0 and 60 nM). After 48 hrs, the cells were collected and fixed for FACS analyses as described above. All data were analyzed using GraphPad Prism (Version 9). Unpaired *t* tests were performed to determine the significance of the readings in respective experimental set up relative to the untreated controls. The data were presented as means ± S.D. of triplicates, whereby *p*-value $< 0.05$ is represented as *; *p-value* $< 0.001$ is represented as ***; *ns* indicates not significant.

### *In vivo* pharmacokinetics

Study design: Twelve SD rats were divided into four groups with 3 animals/group (Study no. KRICT-2014102701-RPK). Group 1 was dosed with STP0404 at 1 mg/kg by single intravenous administration. Groups 2 to 4 were dosed with STP0404 at 20, 40 and 80 mg/kg, respectively, by single oral administration. The vehicle used for IV study was DMSO: PEG400: Water (5,50:45, v:v:v) and the vehicle used for oral studies was 0.5% (w/v) methylcellulose (MC) in water. Blood samples were collected at 0.0833, 0.25, 0.5, 1, 2, 4, 8, 12 (only for PO groups) and 24 hrs post-dose for Groups 1 to 4. STP0404 concentrations in plasma samples were determined by LC-MS/MS. Preclinical Dog pharmacokinetics study (Study no. 400751-2015020201-DPK): A total of 6 male Beagle dogs were divided into two groups (3 animals/group). Following single STP0404 2 mg/kg oral and intravenous administration to Beagle Dogs, STP0404 concentration was determined in plasma, and pharmacokinetic parameters were calculated. Blood samples were collected at 0.033 (for only IV group), 0.0833, 0.25, 0.5, 1.5, 3, 5, 8, 12 and 24 hrs post dose. The concentrations of STP0404 in plasma were determined by using a LC/MS/MS method.

### Safety pharmacology (GLP)

Effects on the Respiratory System in Rats (Study no. YL18404). Four groups of eight male rats were given a single oral administration of STP0404 at doses of 100, 300 and 600 mg/kg. STP0404 was administered as a suspension in 0.5% MC in a volume of 10 mL/kg. Eight control males received the vehicle, 0.5 w/v% methylcellulose solutions, in a similar manner. Measurements were conducted at pre-dosing and 0.5, 1, 3, 6 12 and 24 hrs post-dosing, and respiratory rate, tidal volume and minute volume were investigated (INA Research, Japan). Effects on the Central Nervous System in Rats (Study no. YL18405). Four groups of six male rats were given a single oral administration of STP0404 at doses of 0 (vehicle), 100, 300 and 600 mg/kg. The test article was administered as a suspension in 0.5% MC in a volume of 10 mL/kg. Six control males received the vehicle, 0.5 w/v% methylcellulose solutions, in a similar manner. Blinded observations were conducted pre-dosing and 0.5, 1, 3, 6, 12 and 24 hrs post-dosing on rats body temperature, pupil size, landing foot-splay and grip strength in functional observational battery. Effects on the Cardiovascular System in Dogs (Study no. YL18406). STP0404 at dose levels of 30, 60 and 90 mg/kg were administered orally (by capsules) to four male Beagle dogs in a design with 7-day intervals between doses. Heart rate, blood pressure (systolic, diastolic and mean) and electrocardiographic (ECG) parameters (PR, QT and QTc intervals and QRS duration) were evaluated at pre-dosing and 0.5, 1, 2, 4, 8 and 24 hrs post-dosing. Control animals received empty gelatin capsules in a similar manner for comparison. This test was conducted by a fee to service through INA Research (INA Research, Japan).

### Genotoxicity (GLP)

Bacterial reverse mutation assay (Study no. YL18407). The bacterial reverse mutation assay of STP0404 was performed in two conditions (with or without metabolic activation) for the dose range-finding test (doses of STP0404rangingfrom 2.29 to 5,000μg/plate) and the main test (doses of STP0404 ranging from 19.5 to 5,000 μg/plate) in comparison with negative control (DMSO) and active controls (sodium azide, 2-(2-furyl)-3-(5-nitro-2-furyl) acrylamide, 2-aminoanthracene, 9-aminoacridine hydrochloride). *Salmonella typhimurium* (TA100, TA1535) and *Escherichia coli* (WP2*uvr*A) were used for detection of base-pair substitutions. *Salmonella typhimurium* (TA98, TA1537) was used for detection of frameshift mutations. Chromosomal Aberration Test of STP0404 in Cultured Mammalian Cells (Study no. YL18408). Presence/absence of genotoxicity of STP0404 was determined using chromosomal aberration test

carried out in CHL/IU cells. The test comprised a dose range-finding test and a main test. Micronucleus Study of STP0404 by oral administration in Rats (Study no. YL18409). STP0404 was administered orally to SD rats (3/group in preliminary test and 6/group in the main test) at dose levels of 500, 1000 and 2000 mg/kg/day once daily for 2 days in a two-test study (preliminary test and main test) to investigate the genotoxicity profile of STP0404. Clinical observations and body weight changes were documented. Bone marrow smear slides were evaluated (INA Research, Japan).

## Toxicity (GLP)

STP0404 was administered orally to 10 or 15 SD rats/sex/group at dose levels of 100, 300 and 600 mg/kg/day for 4 weeks to evaluate its potential toxicity. The reversibility of any effects was also assessed following a 2-week untreated recovery period. Control animals (15 animals/sex) received the vehicle, 0.5 w/v% methylcellulose solution, in a similar manner for comparison. In addition, plasma STP0404 concentrations were determined using TK satellite animals (3 animals/sex/group) to evaluate systemic exposure of the animals to the test article. (Study no. YL18402). STP0404 was administered orally as a capsule to 4 or 6 dogs/sex/group at dose levels of 30, 60 and 90 mg/kg/day for 4 weeks to evaluate its potential toxicity. Control animals (6 animals/sex) received empty gelatin capsules in a similar manner for comparison. The reversibility of any effects was also assessed following a 2-week untreated recovery period (2 animals/sex/group for the control and 90 mg/kg/day groups). In addition, plasma STP0404 concentrations were determined using all tested animals (including control group) to evaluate systemic exposure of the animals to the test article (Study no. YL18403). The test was performed according to the Standard Operating Procedures (SOP) the Good Laboratory Practice (GLP) system of the INA Research.

## Microsomal stability determination

A liver microsome (LM) stability assay was six-time points of incubation at 0, 10, 20, 30 and 60 min with a 1 μL STP0404 initial concentration. All plates were shaken and centrifuged at 3200 x g for 20 mins. Then 100 μL of supernatant was taken from each well and diluted with 300 μL pure water before analyzed by LC/MS/MS. Animal and human liver microsomes were purchased from Wuxi AppTec, Xenotech or Corning and stored in a freezer (lower than -60°C) before use (Wuxi AppTec, China).

## Plasma stability determination

STP0404 was incubated with human, monkey, dog, rat and mouse plasma. These incubations were carried out at a test concentration of 5 μM with an incubation period of 60 mins. Samples of human, monkey, dog, rat and mouse were taken at 0, 15, 30, 45, 60 mins. And stop the reaction by taking 50 μL aliquots to 400 μL acetonitrile with internal standard. Propantheline was used as positive control for human, monkey and mouse plasma and mevinolin as the positive control for dog and rat plasma. The remaining percentage was tested. This test was conducted by a fee to service through Pharmaron (Beijing, China).

## Hepatocyte stability determination

Incubation was conducted in 96-well cell culture plate format. The plates were labeled as T0, T15, T30, T60, T90, T0-MC, T90-MC and Blank, respectively. All plates were pre-warmed for 10 to 20 mins at 37±1°C in an incubator of 5% $CO_2$ and saturated humidity. The vials of cryopreserved SD rat, beagle dog, cynomolgus monkey and human hepatocyte were removed from the liquid nitrogen container and immediately immersed in a water bath (the setting

temperature was 37˚C) for approximately 90 secs to allow the ice pellets melted. The melted ice pellets were transferred into pre-warmed 40 mL of Thawing Medium tubes mixed well by gently inverting the tubes then centrifuged at 100 ×g for 5 mins at room temperature. The supernatants were discarded and cell pellets were re-suspended by adding appropriate volumes of pre-warmed WEM. The cells viability of each species was determined using Trypan Blue exclusion. The viabilities of rat, dog, cynomolgus and human hepatocytes were 89.0%, 94.3%, 88.0% and 91.8%, respectively. Cells were finally diluted to $0.625 \times 10^6$ cells/mL with pre-warmed WEM. At each corresponding time point, the sample reactions were stopped by adding 150 µL of acetonitrile containing 200 ng/mL tolbutamide and 200 ng/mL labetalol as internal standards. All sample plates were thoroughly mixed and centrifuged at 3220 ×g for 20 mins. The supernatants were diluted at ratio of 1:3 with ultra-pure water for STP0404 and control samples then submitted for LC/MS/MS analysis (Wuxi AppTec, China).

## Plasma protein binding

The current study utilized the 96-well equilibrium dialysis device purchased from HTDialysis to dialyze the test compound and the control compound at 2 µM in plasma at 37±1˚C for 4 hrs. Plasma prepared from Sprague-Dawley rats, beagle dog, cynomolgus monkey and humans were previously frozen at below -60˚C. The dialysis membrane strips were soaked in ultra-pure water at room temperature for approximately 1 hr. After that, each membrane strip that contains 2 membranes was separated and soaked in ethanol:water (20,80 v,v) for approximately 20 mins, after which it was ready for use or was stored in the solution at 2–8˚C for up to a month. Prior to the experiment, the membrane was rinsed and soaked for 20 mins in ultra-pure water. On the day of experiment, the plasma from Sprague-Dawley rats, beagle dogs, cynomolgus monkey and humans was thawed under running cold tap water and centrifuged at 3220×g for 5 mins to remove any clots and the pH value of the resulting plasma will be checked. if required, adjusted to 7.4 ± 0.1. All samples were further processed by protein precipitation for LC/MS/MS analysis (Wuxi AppTec, China).

## Cytochrome P450 inhibition assay

The final concentrations of test article in incubation system were 0, 0.15, 0.5, 1.5, 5, 15 and 50 µM. The inhibition of CYP1A2, 2B6, 2C8, 2C9, 2C19, 2D6, and 3A4 in human liver microsomes was measured as the percentage decrease in the activity of marker metabolite formation compared to non-inhibited controls (= 100% activity). The rate of formation of isoform-specific metabolites derived from marker substrates were measured by UPLC-MS/MS. The $IC_{50}$ values were calculated with the metabolites decrease percentages of test samples to vehicle samples. All incubations were performed in triplicate (Pharmaron, China).

## PXR Induction

STP0404 to activate PXR, and thereby to induce CYP3A4 expression in DPX2 cell lines following in vitro administration. DPX2 cells were treated with STP0404 at 0.0977, 0.391, 1.56, 6.25, 25.0 and 100 µM at 37˚C for a total of 48 hrs with the change of incubation medium every 24 hrs. After 48 hrs incubation, the cell viability was assessed fluorimetrically via CellTiter-FluorTM. CYP3A-mediated metabolism and PXR activation are measured using luminescence with Luciferin-IPA and ONE-GloTM, respectively. The light intensity is directly proportional to the extent of PXR activation and accompanying gene transcription in the DPX2 cells (Wuxi AppTec, China).

## Cell cytotoxicity assay

STP0404 was tested in L929, HFL-1, NIH3T3, VERO and CHO-K1 cells using CellTiter-Glo Luminescent Cell Viability Assay kit, in 96-well assay plates. IC$_{50}$ values of positive control Staurosporine were 0.002 μM, 0.0004 μM, 0.0004 μM, 0.002 μM, and 0.0021 μM in L929, HFL-1, NIH3T3, CHO-K1 and VERO cells, respectively (Wuxi AppTec, China).

## Supporting information

**S1 Table. Data collection and refinement statistics.**
(PPTX)

**S2 Table.** *In vitro* **ADME profile of STP0404.**
(PPTX)

**S3 Table.** *In vivo* **safety pharmacology and repeat toxicity of STP0404.**
(PPTX)

**S1 Fig.** *In vitro* **efficacy tests of STP0404, BI224436 and Raltegravir against SIVmac239 in CEMx174 cell culture.** CEMx174 cells were infected with SIVmac239 with various concentrations of STP0404, BI224436, and Raltegravir, and the viral production was determined by p27 assay at 5 days post infection.
(TIFF)

**S2 Fig. Close up image of STP0404 binding of the LEDGF/p75 binding pocket of IN dimer.** This is a close up image of the STP0404 binding (mesh) to the LEDGF/p75 binding pocket of IN dimer shown in Fig 3D.
(TIFF)

## Author Contributions

**Conceptualization:** Kyungjin Kim, Baek Kim.

**Data curation:** Tatsuya Maehigashi, Seohyun Ahn, Uk-Il Kim, Jared Lindenberger, Adrian Oo, Pratibha C. Koneru, Bijan Mahboubi.

**Formal analysis:** Tatsuya Maehigashi, Seohyun Ahn, Uk-Il Kim, Jared Lindenberger, Adrian Oo, Pratibha C. Koneru, Bijan Mahboubi.

**Funding acquisition:** Mamuka Kvaratskhelia, Kyungjin Kim, Baek Kim.

**Investigation:** Mamuka Kvaratskhelia, Kyungjin Kim, Baek Kim.

**Methodology:** Seohyun Ahn, Uk-Il Kim, Alan N. Engelman.

**Project administration:** Baek Kim.

**Resources:** Alan N. Engelman.

**Supervision:** Mamuka Kvaratskhelia, Kyungjin Kim, Baek Kim.

**Validation:** Tatsuya Maehigashi, Seohyun Ahn.

**Writing – original draft:** Baek Kim.

**Writing – review & editing:** Seohyun Ahn, Uk-Il Kim, Adrian Oo, Mamuka Kvaratskhelia, Kyungjin Kim.

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
