## [Decision Letter · Decision Letter 0]

1 Apr 2021

Dear Dr. Kim,

Thank you very much for submitting your manuscript "A Highly Potent and Safe Pyrrolopyridine-based Allosteric HIV-1 Integrase Inhibitor Targeting Host LEDGF/p75-integrase Interaction Site" for consideration at PLOS Pathogens. As with all papers reviewed by the journal, your manuscript was reviewed by members of the editorial board and by three independent reviewers. The reviewers appreciated the attention to an important topic. Based on the reviews, we are likely to accept this manuscript for publication, providing that you modify the manuscript according to the review recommendations.

Sincerely,

Bryan R. Cullen

Associate Editor

PLOS Pathogens

Thomas Hope

Section Editor

PLOS Pathogens

Kasturi Haldar

Editor-in-Chief

PLOS Pathogens

orcid.org/0000-0001-5065-158X

Michael Malim

Editor-in-Chief

PLOS Pathogens

orcid.org/0000-0002-7699-2064

Reviewer Comments (if any, and for reference):

Reviewer's Responses to Questions

**Part I - Summary**

Reviewer #1: This is a beautiful paper that describes the first potentially clinically useful allosteric HIV-1 integrase inhibitor, a very impressive achievement. I only have several very minor comments.

Reviewer #2: In this study, the authors report on the discovery and characterization of a new pyrrolopyridine-based scaffold ALLINI, STP0404. Integrase has been efficiently targeted by numerous compounds that target its catalytic activity. ALLINIs are a promising class of compounds that target the interaction between LEDGF and IN (early stage of infection), but more potently IN-RNA interactions during virion maturation (late stage of infection). In doing so, ALLINIs interfere with proper virion maturation resulting in the formation of aberrant virions with eccentrically localized viral genomes. Despite the promise of ALLINIs, most compounds published to date display unfavorable profiles in vivo and as a result did not advance to use in humans. The major advancement of this manuscript is the fact that compared with other ALLINIs STP0404 shows outstanding therapeutic and safety properties. The remainder of the findings are in line with what is already known about ALLINIs, though I thank the authors for conducting numerous experiments validating this compound carefully. I think the manuscript is well suited for Plos Pathogens and well written but I would address a few points increase the impact.

Reviewer #3: This is an exciting paper that reports the properties of an HIV-1 inhibitor that targets the LEDGF binding site of HIV-1 integrase. Although such inhibitors (ALLINIs) have been previously described, this compound (STP0404) is the first example of such a compound that meets the necessary criteria, from cell based assays and animal studies, to enter human clinical trials. At minimum, the work is proof of concept that that this new class of HIV-1 inhibitors are viable drug candidates. Futhermore, STP0404 has the highly desirable property that resistance mutants in HIV-1 integrase exhibit impaired replication. This is a very significant paper that will stimulate the development of this new class of drugs for the treatment of HIV-1. I recommend it be accepted for publication in PLOS Pathogens after a couple of minor issues have been addressed.

**Part II – Major Issues: Key Experiments Required for Acceptance**

Reviewer #1: No major issues

Reviewer #2: 1- Given the big reduction in fitness with the Y99H substitution, the authors should probably have passaged this virus further in the presence of STP0404 to gain full resistance. Alternatively they can clone in previously described resistance mutations and see whether they more potently restore virion infectivity in the absence/presence of the compound.

2- For the in vitro PK assays, I would love to see a comparison with 1-2 most potent ALLINIs.

3- The in vivo PK studies are great but they fail to address one point which is whether the compound is effective against reducing virus titers in vivo. I think it would be important to show this in a humanized mouse model if at all possible.

Reviewer #3: None

**Part III – Minor Issues: Editorial and Data Presentation Modifications**

Reviewer #1: 1) Figure 4b: How many virions were imaged? Methods (page 16) should include at least very brief details of the staining and imaging procedures.

2) Inhibition of IN-LEDGF interaction (page 8): I suggest highlighting that the assay used full-length HIV-1 IN. It is impressive that the small molecule can do that.

3) Is the compound active (or is not expected to be active) against HIV-2? Can we expect issues with some HIV-1 clades? I am not suggesting adding data; a few words on this would improve Discussion.

4) An image of electron density of the drug-bound pocket should be included.

Reviewer #2: Perhaps intro and discussion can be expanded a little bit to compare the compound to other ALLINIs and why the authors think it is better.

Reviewer #3: 1. "Low nanomolar" would appear to be a more accurate description of the IC50 range than "picomolar".

2. While the data in Table 1 support the authors claim that STP0404 is highly potent in both wild-type and Ral-resistant HIV-1 strains, the reader is left wondering why the STP0404 IC50s are considerably lower in most of the Ral-resistant strains. Is this simply due to other differences in these strains compared to the Nl4-3 comparator? Some comment would be helpful.

PLOS authors have the option to publish the peer review history of their article (what does this mean?). If published, this will include your full peer review and any attached files.

Reviewer #1: No

Reviewer #2: No

Reviewer #3: **Yes: **Robert Craigie

Figure Files:

Data Requirements:

Reproducibility:

References:

---

## [Editor Report · Decision Letter 1]

27 May 2021

Dear Dr. Kim,

We are pleased to inform you that your manuscript 'A Highly Potent and Safe Pyrrolopyridine-based Allosteric HIV-1 Integrase Inhibitor Targeting Host LEDGF/p75-integrase Interaction Site' has been provisionally accepted for publication in PLOS Pathogens.

Best regards,

Bryan R. Cullen

Associate Editor

PLOS Pathogens

Thomas Hope

Section Editor

PLOS Pathogens

Kasturi Haldar

Editor-in-Chief

PLOS Pathogens

orcid.org/0000-0001-5065-158X

Michael Malim

Editor-in-Chief

PLOS Pathogens

orcid.org/0000-0002-7699-2064
---

## [Editor Report · Acceptance letter]

28 Jun 2021

Dear Dr. Kim,

We are delighted to inform you that your manuscript, "A Highly Potent and Safe Pyrrolopyridine-based Allosteric HIV-1 Integrase Inhibitor Targeting Host LEDGF/p75-integrase Interaction Site," has been formally accepted for publication in PLOS Pathogens.

Best regards,

Kasturi Haldar

Editor-in-Chief

PLOS Pathogens

orcid.org/0000-0001-5065-158X

Michael Malim

Editor-in-Chief

PLOS Pathogens

orcid.org/0000-0002-7699-2064